# Temporal Trends in Syphilis Incidence among Men with HIV in Busan, Korea, 2005–2022: A Retrospective Cohort Study

**DOI:** 10.3390/v16020265

**Published:** 2024-02-07

**Authors:** Sun Hee Lee, Jeong Eun Lee, Soon Ok Lee, Shinwon Lee, Woo Seog Ko, Hyung-Hoi Kim, Kyung-Hwa Shin, Jin Suk Kang, Hyunjin Son

**Affiliations:** 1Department of Internal Medicine, Pusan National University School of Medicine and Pusan National University Hospital, Busan 49241, Republic of Korea; zzanmery@gmail.com (S.H.L.); godprayer166@naver.com (J.E.L.); lsook81@hanmail.net (S.O.L.); plusmed2143@gmail.com (W.S.K.); 2Department of Laboratory Medicine, Pusan National University School of Medicine, Busan 49241, Republic of Korea; hhkim@pusan.ac.kr (H.-H.K.); skyoungh@naver.com (K.-H.S.); 3Biomedical Informatics Unit, Pusan National University School of Medicine, Busan 49241, Republic of Korea; 4Department of Internal Medicine, Inje University College of Medicine, Busan 47392, Republic of Korea; gmlsenddl06@naver.com; 5Department of Prevention Medicine, Donga University School of Medicine, Donga University Hospital, Busan 49201, Republic of Korea; hjson@dau.ac.kr

**Keywords:** syphilis, incidence, HIV, sexually transmitted disease

## Abstract

We aimed to assess the temporal trends of incident syphilis and its associated risk factors among men with HIV (Human Immunodeficiency Virus) in Korea during the COVID-19 pandemic. We conducted a retrospective cohort study of men with HIV attending an HIV clinic in Korea between 2005 and 2022. Of 767 men with HIV, 499 were included and contributed 3220 person-years (PY) of the observation period. Eighty-two patients were diagnosed with incident syphilis, with an overall incidence of 2.55/100 PY (95% confidence interval [CI] 20.56–31.53). The incidence of syphilis per 100 PY gradually decreased from 2.43 (0.79–7.42) in 2005–2007 to 1.85 (1.08–3.17) in 2014–2016; however, it increased to 3.0 (1.99–4.53) in 2017–2019, and further to 3.33 (2.26–4.89) in 2020–2022. A multivariate analysis identified young age (≤30 years versus >50, adjusted HR 6.27, 95% CI 2.38–16.56, *p* < 0.001), treponemal test positive at baseline (2.33, 1.48–3.67, *p* < 0.001), men who have sex with men (2.36, 1.34–4.16, *p* = 0.003), and history of incarceration (2.62, 1.21–5.67, *p* = 0.015) as risk factors for incident syphilis. Recently, syphilis incidence in men with HIV has increased in Korea, especially in young patients and at-risk groups, highlighting the need for enhanced regular screening and targeted behavioral interventions among these populations.

## 1. Introduction

Globally, the incidence of syphilis has steadily risen over the past decade, with wide variations among countries and regions depending on their socioeconomic status [1]. In higher-income countries, syphilis infection is less common and occurs disproportionately in racial, ethnic, and sexual minority populations [2]. However, recently, an unexpected increase in syphilis has been seen, particularly in populations that had previously received comparatively less attention or were overlooked [3], such as in women, congenital syphilis [4,5,6,7], and syphilis in older adults [8,9].

In the United States (US), primary and secondary (P&S) syphilis has increased almost every year since reaching a historic low in 2000 and 2001, among both males and females, in all regions of the US and all age groups [10]. Men who have sex with men (MSM) are disproportionately impacted by syphilis, accounting for almost half of all male P&S syphilis cases in 2021. However, the incidence of P&S syphilis among heterosexual individuals and women has increased significantly in recent years. The rate of P&S syphilis increased by 24% among reproductive-aged women from 2019 to 2020, resulting in an increase in congenital syphilis in the US [10].

In Japan, immediately after the emergence of the COVID-19 pandemic, the number of reported syphilis cases decreased but eventually reversed, showing significantly higher reporting in mid-to-late 2021, mainly in heterosexual men and women [11,12]. However, the incidence of syphilis in individuals with HIV decreased, suggesting that syphilis in individuals with HIV did not contribute to the recent syphilis resurgence in Japan [13].

Both syphilis and Human Immunodeficiency Virus (HIV) are transmitted sexually, and coinfection is common. Infection with one may increase the acquisition and transmission of the other [14]. The HIV–syphilis coinfection rate varies depending on the prevalence of each infection in the community and individual risk factors [15]. Asymptomatic syphilis is also commonly reported in patients with HIV, and frequent screening for syphilis as part of HIV monitoring can increase the early detection of asymptomatic syphilis [16,17,18,19]. Incident syphilis can be an indicator of unprotected sexual contact and is associated with incident HIV infection. Therefore, continuous monitoring of syphilis incidence in individuals with HIV can help understand the latest changes in syphilis epidemiology and provide insights into prevention strategies for the population of individuals with HIV.

In Korea, the number of reported cases of P&S syphilis in the general population increased from 1015 in 2014 to 2270 in 2018, and the proportion of males with syphilis increased from 56% in 2014 to 71.2% in 2018 [20]. Only a few studies have reported syphilis epidemiology in individuals with HIV in Korea [21,22,23]. In a single-site study of 539 patients with HIV undergoing anti-retroviral therapy (ART) from 1998 to 2006, the incidence rate of early syphilis 4 years after starting ART was 4.57 per 100 person-years [23]. All 56 patients diagnosed with early syphilis were male. In another study using health insurance claims data collected between 2008 and 2016, 48.3% of 9393 individuals with HIV who underwent ART were diagnosed with syphilis coinfection, which was defined as a syphilis-related diagnosis with a benzathine penicillin G prescription on the same day, and 94.3% of those with syphilis coinfection were male [22].

However, no studies have been conducted on the incidence of syphilis among individuals with HIV since 2016 in Korea. Therefore, it is important to provide an update on the changes in the epidemiology of syphilis during the COVID-19 era. In Korea, 93.6% of individuals with HIV are male, and the majority of reported cases of syphilis in patients with HIV are men. Thus, we aimed to examine the temporal trends of incident syphilis and the associated risk factors among men with HIV from 2005 to 2022.

## 2. Materials and Methods

### 2.1. Study Design and Definition

This retrospective cohort study was conducted at the Pusan National University Hospital (PNUH), which aimed to assess the incidence of syphilis in individuals with HIV. PNUH is a 1220-bed university-affiliated teaching hospital that provides HIV care to individuals with HIV in southeastern Korea. Syphilis serological testing was routinely performed upon entry into care. Subsequently, patients are typically followed up at 3–6 monthly intervals, and their rapid plasma reagin (RPR) titers are assessed every 6 months. We included men with HIV who visited the study hospital for the first time between 2005 and 2022, were aged 18 years or older, had both non-treponemal (non-TP) and treponemal (TP) test results at baseline, and had at least one subsequent RPR titer during the study period. We excluded patients who visited the hospital for reasons other than HIV care and those with prevalent syphilis at baseline (symptomatic or RPR titer ≥ 8) [24,25]. We collected sociodemographic, behavioral, and clinical data, including laboratory and treatment information, through a retrospective review of medical records. We defined incident cases of syphilis as experiencing a 4-fold increase in the RPR titer if the baseline TP test was positive or new seroreactivity (positive non-TP and TP tests) with a previous negative result [26].

Non-TP testing consisted of the RPR card test (Asanpharma, Seoul, Republic of Korea) and RPR turbidimetric immunoassay (TIA, Sekisui Chemical Co., Ltd., Osaka, Japan), and TP testing included T. pallidum hemagglutination assay (TPHA, Asanpharma, Seoul, Republic of Korea), Fluorescent treponemal antibody absorption (FTA-ABS, ZEUS Scientific, Branchburg, NJ, USA), T. pallidum particle agglutination (TPPA, Fujirebio, Tokyo, Japan), and Treponema pallidum latex agglutination (TPLA, Sekisui Chemical Co., Ltd., Osaka, Japan).

Stages of syphilis were determined by an infectious disease specialist according to the US Centers for Disease Control and Prevention’s (CDC) surveillance definitions [27]. Latent syphilis acquired within the preceding year was referred to as early latent syphilis, and all other cases of latent syphilis were classified as late latent syphilis or latent syphilis of unknown duration. Illicit drugs were defined as substances that were legally prohibited in Korea under the Act on the Control of Narcotics (narcotics, psychotropic drugs or substances, cannabis, and some substances designated as temporary narcotics, such as nitrite inhalants) and other substances (sniffing of glue) prohibited by the Act on the Protection of Children and Youth against Sex Offenses and the Chemical Substance Control Act [28].

### 2.2. Statistical Analysis

The observation period for individual patients was from the first visit to the study hospital to the earliest of the following dates: date of incident syphilis diagnosis, death, transfer to another institute, or 31 December 2022. To evaluate the trends in the incidence of syphilis, we divided the overall observation period into six calendar periods (2005–2007, 2008–2010, 2011–2013, 2014–2016, 2017–2019, and 2020–2022). The incidence of syphilis was calculated by dividing the number of incident syphilis cases by person-time at risk and was expressed per 100 person-years (PY). The 95% confidence intervals (CIs) of incidence were calculated using Poisson distribution. Syphilis testing rates per year were determined based on the presence of any non-treponemal (diagnostic or screening) syphilis test in each given calendar year when a patient was under observation [26,29].

Categorical variables were compared using Pearson’s *χ*^2^ test or Fisher’s exact test, whereas non-categorical variables were tested using a *t*-test. The time from the first visit to the incident syphilis diagnosis was assessed using Kaplan–Meier curves. A Cox regression analysis was conducted to identify risk factors for incident syphilis. All statistical analyses were performed using R version 4.3.1 (R Foundation for Statistical Computing, Vienna, Austria). All tests were two-tailed, and a *p*-value of <0.05 was considered significant.

## 3. Results

Among the 953 men with HIV who visited the study hospital for the first time between 2005 and 2022, 454 were excluded from the analysis of the incidence of syphilis for the following reasons: 53 did not undergo any syphilis tests, 133 had only one syphilis test, 8 visited the hospital for reasons other than HIV care, 114 had prevalent syphilis at baseline, and 146 did not undergo a follow-up RPR titer test (Figure 1). Finally, 499 men with HIV were included, contributing 3220 PY during the observation period.

The median patient age was 41 years (interquartile range [IQR, 31–51]). Most patients (97.4%) were ethnic Koreans, and 61.7% were unmarried. More than half (57.7%) were MSM, and 85.4% were ART-naïve. The Median CD4 T cell count was 262/µL (IQR 100–417), and 34.7% were TP test-positive at baseline. The median testing rate was 2.2 per year. The median observation period per patient was 5.08 years (IQR 1.97–9.930) (Table 1).

A total of 82 patients were diagnosed with incident syphilis, with an overall incidence of 2.55/100 PY (95% CI 20.56–31.53). The median time from the first visit to the incident syphilis was 3.03 years (IQR 1.01–6.9). The incidence of syphilis per 100 PY gradually decreased from 2.43 (0.79–7.42) in 2005–2007 to 1.85 (1.08–3.17) in 2014–2016, but recently increased to 3.0 (1.99–4.53) in 2017–2019, and further to 3.33 (2.26–4.89) in 2020–2022 (Figure 2).

Overall, 90.2% (74/82) of the patients with incident syphilis were diagnosed during the early syphilis stage. More than half of the incident syphilis episodes (46/82, 51.6%) occurred during the early symptomatic syphilis stage, and the most common features were skin rash (39/82, 47.6%), anogenital lesions (9/82, 11%), and oropharyngeal lesions (3/82, 3.7%). Nearly one-third of the cases (28/82, 34.2%) were detected during the asymptomatic early latent stage. Seven patients (8.5%) were classified as having late latent syphilis or latent syphilis of unknown duration. Asymptomatic neurosyphilis was observed in one episode (1.2%).

Compared to men with HIV without incident syphilis, those with incident syphilis were more likely to be younger (median age 34 vs. 42 years), unmarried (78% vs. 58.5%), MSM (78.1% vs. 53.7%), and to have positive TP test results at baseline (50% vs. 31.7%) (Table 1). Kaplan–Meier analyses showed that the cumulative infection rate after enrollment was significantly higher in the younger age group (*p* < 0.01), MSM group (*p* < 0.01), and TP test positive at baseline group (*p* < 0.01) (Figure 3). In the univariate Cox regression analysis, younger age, unmarried status, MSM, and positive TP test results at baseline were associated with a higher risk of incident syphilis (Table 2). A multivariate Cox regression analysis identified young age (≤30 years versus >50, adjusted HR 6.27, 95% CI 2.38–16.56, *p* < 0.001), TP test positive at baseline (2.33, 1.48–3.67, *p* < 0.001), MSM (2.36, 1.34–4.16, *p* = 0.003), and history of incarceration (2.62, 1.21–5.67, *p* = 0.015), as risk factors for incident syphilis (Table 2).

## 4. Discussion

During the COVID-19 era, newly diagnosed HIV cases in Korea decreased from 1223 in 2019 to 1016 in 2020, 975 in 2021, and increased again to 1066 in 2022 [30]. The decline observed between 2020 and 2021 seemed to be due to the impact of COVID-19-related restrictions on testing behavior.

During 2020–2021, public health centers (PHCs) temporarily suspended HIV tests, and their resources were disproportionately allocated to respond to the COVID-19 outbreak. The proportion of confirmed cases of HIV through testing at PHCs dropped steeply from 30.0% in 2019 to 16.3% in 2020 and 16.1% in 2021 [30,31].

In Korea, syphilis is managed and regularly monitored for P&S and congenital syphilis in high-risk groups for sentinel surveillance. However, this was changed to mandatory universal surveillance in 2011 and again to sentinel surveillance in 2020 [32,33]. Reported cases of P&S and congenital syphilis in the general population have increased steadily since 2001, peaked in 2008, decreased until 2012, started to increase again in 2014 (1015 cases), increased rapidly since 2016 (1568 cases), reached a new peak in 2018 (2280 cases), and decreased again in 2019 (1753 cases) [20,33]. Since mandatory surveillance for syphilis was changed to sentinel surveillance in 2020, it is difficult to directly compare the reported cases under sentinel surveillance with the number of reported cases, which also steadily increased in the COVID-19 era, with 330 cases in 2020, 390 cases in 2020, 2021, and 401 cases in 2022 [30].

In our study, the incidence of syphilis per 100 PY in patients with HIV gradually decreased from 2.43 in 2005–2007 to 1.85 in 2014–2016 but recently increased to 3.0 in 2017–2019, and further to 3.33 in 2020–2022, the COVID-19 era. These temporal trends are similar to those observed in the general Korean population. However, it is unclear whether individuals with HIV contributed to the recent increase in syphilis in the general population and how the disruption of HIV testing in PHCs during the COVID-19 outbreak could affect HIV spread in Korea, and further research is needed.

In Korea, the proportion of individuals with HIV is less than 0.1% of the total population. Therefore, it is difficult to conclude that those infected with HIV are leading the overall syphilis trend. Rather, the continuous increase in syphilis incidence among individuals with HIV during the COVID-19 pandemic may be a reflection of this increase in syphilis incidence in the general population since 2016 [33]. However, the reasons for the recent increase in the prevalence of syphilis in patients with HIV in Korea may be multifactorial. In our study, compared to men who did not have incident syphilis during the observation period, those who had incident syphilis were more likely to be younger, MSM, have a positive TP test at baseline, and have a history of incarceration.

Our study showed a higher risk of acquiring syphilis among young men aged ≤30 years compared with those >50 years (5.69 vs. 0.98 per 100 PY). These trends have also been observed in previous studies of men with HIV [25,34,35]. On a national scale, the proportion of individuals with HIV aged <30 years has gradually increased from 23% in 2008–2010 to 31.8% in 2011–2013 and 36.9% in 2014–2016 and has since fluctuated but tended to be between 34 and 39% in Korea [30]. In our study, the proportion of men aged ≤30 among newly enrolled patients has also steadily increased from 13.5% in 2008–2010 to 29.5% in 2014–2016 and 46.8% in 2020–2022. The continued proportional age shift in patients with HIV could have contributed to the recent increase in syphilis cases in this study.

Since 2019, homosexual contact has been listed as the most common route of HIV infection in the annual report issued by the Korea Centers for Disease Control and Prevention (KCDC), based on epidemiological investigations conducted by PHCs for newly diagnosed individuals with HIV [30]. In a previous study conducted between 2003 and 2014 in five Asian countries, including Korea, the incidence of syphilis was higher in MSM than in non-MSM (7.64 vs. 2.44 per 100 PY) [21]. In our study, the incidence of syphilis was more than two times higher among MSM compared with non-MSM (3.4 vs. 1.27 per 100 PY). Overall, 58.1% of patients were MSM. Younger men were more likely to be MSM, with 71.3% of those aged ≤ 30 years, 58.2% of those aged 31–50 years, and 45.2% of those older than 50 years, suggesting that more targeted efforts for young MSM are needed.

In our study, men with positive TP test results at baseline were more likely to have incident syphilis than those with negative TP tests (3.85 vs. 1.9 per 100 PY). Among the 172 patients with positive TP test results at baseline, 113 (65.7%) had a self-reported history of syphilis treatment, and 110 (58.1%) had discordant results at baseline with a negative non-TP test. TP tests usually remain positive after prior treatment of infection, although positivity may occur in patients treated in the primary stage of syphilis and in those with advanced HIV disease [36,37]. Discordant results may reflect either prior syphilis, false-positive TP tests, or very early or late syphilis. Although the TP test is more complex in patients with a history of syphilis, our study suggests that patients with a history of syphilis are more likely to have incident syphilis than those without a history of syphilis.

In the present study, the incidence of syphilis was higher in patients with a history of incarceration before or during hospital visits than in those without (4.97 vs. 2.42 per 100 PY). The reasons for custody varied and included drug use; however, the reason was not disclosed in many cases. However, there were no significant differences in the incidence of syphilis according to the history of illicit or injected drug use in our study. There are several reasons for these findings, but underreporting may be an important factor [28]. There were some limitations in obtaining accurate data from custody and illicit drug users.

Frequent serologic screening can identify persons recently infected and sometimes before infectious lesions develop [38]. The CDC suggests annual preventive screening for sexually active MSM [39]. However, more frequent screening at 3-month intervals may increase the detection of syphilis [40]. In our study, 90.2% of patients with incident syphilis were diagnosed during the early syphilis stage, of whom 37.8% were detected during the asymptomatic early latent stage. The median test rate was 2.2 tests per year. These findings underscore the importance of regular syphilis screenings as part of routine HIV care and the need for more frequent testing of at-risk groups, including younger age, MSM, history of syphilis infection, and incarceration.

Our findings have several limitations. First, it was a hospital-based retrospective cohort study. The retrospective nature of this study may have limited the collection of epidemiological data. Therefore, the presence of unmeasured confounding factors cannot be ruled out. Second, this study was conducted at a single center, and only a small number of patients with HIV infection were included. Therefore, these results should be generalized to other regions of the country with caution. Third, in our study, only the first incident of syphilis episodes was included; therefore, we did not examine subsequent repeat syphilis episodes. This may have led to an underestimation of the true incidence of syphilis. Fourth, in addition to symptomatic patients at the baseline visit, we excluded those who had RPR titer ≥ 8 at baseline as prevalent syphilis. In this case, there was a possibility of recently treated syphilis with decreased RPR titers. However, we cannot exclude the possibility that some of these patients were in a serofast state, which would have led us to overestimate the prevalence of syphilis.

## 5. Conclusions

Our study revealed a recent increasing trend in the incidence of syphilis among men with HIV in Korea. These temporal trends are similar to those observed in the general Korean population. Most patients with incident syphilis were diagnosed during the early syphilis stage, of whom 37.8% were detected during the asymptomatic early latent stage. Younger age, MSM, history of syphilis infection, and incarceration were identified as risk factors for syphilis infection. These findings highlight the need for enhanced regular screening and targeted behavioral interventions in these populations.

## Figures and Tables

**Figure 1 viruses-16-00265-f001:**
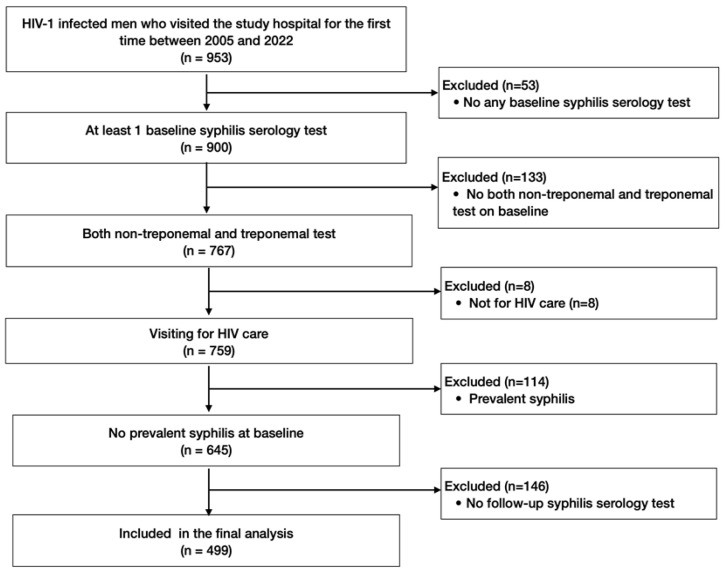
Patient selection flowchart.

**Figure 2 viruses-16-00265-f002:**
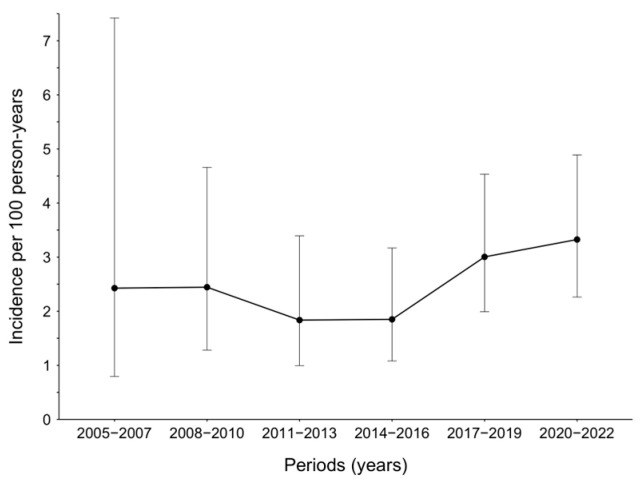
Incidence of syphilis among men with HIV-1, in Busan, Korea, 2005–2022. Upper and lower whiskers represent the 95% confidence interval.

**Figure 3 viruses-16-00265-f003:**
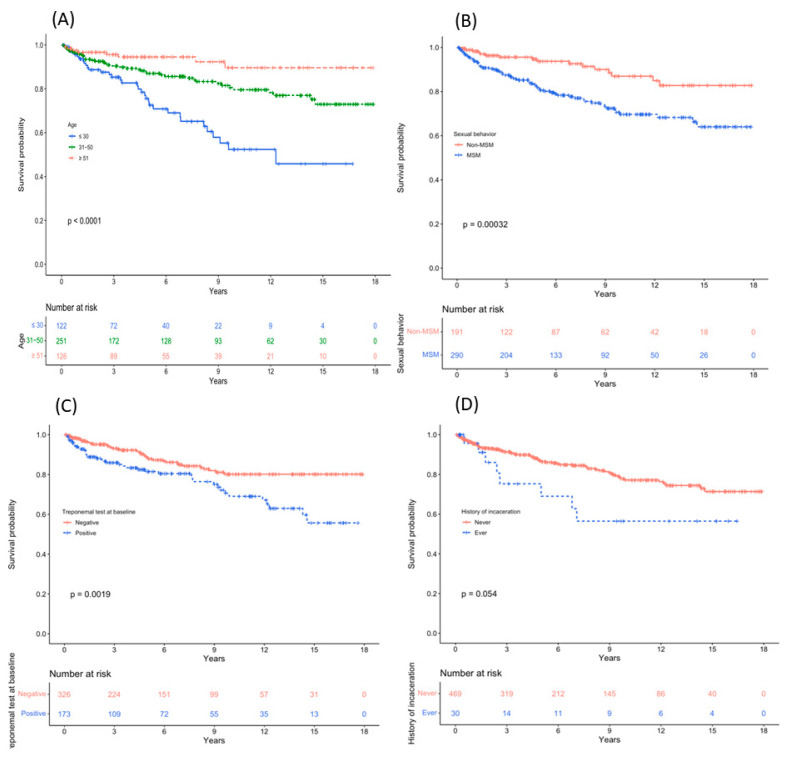
The Kaplan–Meier survival curves of syphilis incidence stratified by age (**A**), sexual behaviors (**B**), treponemal test results at baseline (**C**), and history of incarceration (**D**).

**Table 1 viruses-16-00265-t001:** Baseline characteristics of the study population with and without incident syphilis among men with HIV in Busan, Korea, 2005–2022.

Characteristics	All Patients (n = 499)	Patients with Incident Syphilis (n = 82)	Patients without Incident Syphilis (n = 417)	*p*-Value
Median age at baseline, years [IQR]	41 [31–51]	34 [24–43.3]	42 [32–52]	<0.001
Ethnicity				0.491
Korean	486 (97.4)	81 (98.8)	405 (97.1)	
Non-Korean	13 (2.6)	1 (1.2)	12 (2.9)	
Marriage				0.002
Unmarried	308 (61.7)	64 (78.0)	244 (58.5)	
Married previously	185 (37.1)	18 (22.0)	167 (40.1)	
Unknown	6 (1.2)	0 (0)	6 (1.4)	
Sexual behavior				<0.001
Non-MSM	191 (38.3)	16 (19.5)	175 (42.0)	
MSM	290 (58.1)	65 (79.3)	225 (54.0)	
Unknown	18 (3.6)	1 (1.2)	17 (4.1)	
Median CD4 cell counts, /µL [IQR]	262 [100–417]	302 [150.5–380]	261 [92–422]	0.495
Anti-retroviral therapy				0.393
Naïve	426 (85.4)	73 (89.0)	353 (84.7)	
Experienced	73 (14.6)	9 (11.0)	64 (15.3)	
Treponemal test positive at baseline	172 (34.5)	41 (50.0)	131 (31.5)	0.002
Incaceration, ever	30 (6.0)	8 (9.8)	22 (5.3)	0.128
Illicit drug use, ever	32 (6.4)	8 (9.8)	24 (5.8)	0.213
Injection drug use, ever	14 (2.8)	3 (3.7)	11 (2.6)	0.712
HBV surface antigen, positive	29 (5.8)	3 (3.7)	26 (6.2)	0.449
HCV antibody, positive	16 (3.2)	1 (1.2)	15 (3.6)	0.333
Calendar period of baseline visit				0.289
2005–2007	98 (19.6)	15 (18.3)	83 (19.9)	
2008–2010	96 (19.2)	15 (18.3)	81 (19.4)	
2011–2013	100 (20.0)	21(25.6)	79 (18.9)	
2014–2015	78 (15.6)	16 (19.5)	62 (14.9)	
2016–2019	80 (16.0)	12 (14.6)	68 (16.3)	
2020–2022	47 (9.4)	3 (3.7)	44 (10.6)	
Median testing rates per year [IQR]	2.2 [1.6–2.8]	2.9 [1.7–4.28]	2.2 [1.6–2.7]	0.043
Median follow-up periods, years [IQR]	5.08 [1.97–9.93]	3.03 [1.01–6.9]	5.76 [2.35–11.13]	<0.001

IQR, interquartile range; MSM, male sex; HBV, hepatitis B virus; HCV, hepatitis C virus; Values in parentheses () are percentages (%), unless otherwise indicated.

**Table 2 viruses-16-00265-t002:** Risk factors for incident syphilis for men with HIV in Busan, Korea, 2005–2022.

Variables	Incidences per 100 PY	Univariate Analysis		Multivariate Analysis	
		Crude HR (95% CI)	*p*-Value	Adjusted HR (95% CI)	*p*-Value
Age at enrollment, years					
>50	0.98	Reference			
30–50	2.21	2.3 (1.08–4.92)	0.032	2.35 (0.99–5.58)	0.052
≤30	5.69	5.41 (2.5–11.71)	<0.001	6.27 (2.38–16.56)	<0.001
Ethnicity (non-Korean vs. Korean)	6.14 vs. 2.53	1.62 (0.22–11.92)	0.633		
Marriage (unmarried vs. previously married)	3.57 vs. 1.28	2.63 (1.56–4.44)	<0.001	1.07 (0.57–2.03)	0.825
Sexual behavior (MSM vs. non-MSM)	3.4 vs. 1.27	2.63 (1.52–4.55)	<0.001	2.36 (1.34–4.16)	0.003
CD4 cell counts at baseline, /µL					
≤200	1.97	Reference			
201–500	3.06	1.52 (0.94–2.47)	0.091	1.24 (0.76–2.03)	0.398
>500	2.84	1.39 (0.74–2.61)	0.301	0.99 (0.51–1.91)	0.964
ART (experienced vs. naïve)	2.54 vs. 2.55	0.94 (0.47–1.89)	0.860		
Treponemal test at baseline (positive vs. negative)	3.85 vs. 1.9	1.96 (1.27–3.02)	0.002	2.33 (1.48–3.67)	<0.001
Incarceration (ever vs. never)	4.97 vs. 2.42	2.02 (0.97–4.19)	0.059	2.62 (1.21–5.67)	0.015
Illicit drug use (ever vs. never)	3.11 vs. 2.50	1.26 (0.61–2.62)	0.530		
Injection drug use (ever vs. never)	3.33 vs. 2.52	1.27 (0.40–4.02)	0.684		
HBV surface antigen (positive vs. negative)	1.51 vs. 2.61	0.58 (0.18–1.83)	0.351		
HCV antibody (positive vs. negative)	1.26 vs. 2.58	0.46 (0.06–3.29)	0.437		

PY, person-years; HR, hazard ratio; MSM, men who have sex with men; ART, antiretroviral therapy; HBV, hepatitis B virus; HCV, hepatitis C virus.

## Data Availability

Full data from this study are not available due to the protection of privacy of some data.

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
