# Peer review of "Temporal Trends in Syphilis Incidence among Men with HIV in Busan, Korea, 2005–2022: A Retrospective Cohort Study"

_viruses, 2024, doi:10.3390/v16020265_

Round 1
Reviewer 1 Report
Comments and Suggestions for Authors
The paper of Sun Hee Lee et al. refers to the important problem of syphilis infection among people with AIDS. This is the first study covering this group of patients in Korea since 2016. The study aimed to assess trends in the incidence of syphilis and associated risk factors in men infected with HIV in the years 2005-2022.
The study included men who were first admitted to the care of the Busan National University Hospital in Korea between 2005 and 2022, were at least 18 years of age, and had a treponemal and nontreponemal test performed on admission with at least one additional RPR test during the study period. The period 2005-2022 has been divided into six calendar periods.
Of 953 HIV-infected men, 499 patients were enrolled in the study. Among them, more than half belonged to the MSM group and 85% had not received antiretroviral therapy before admission. HIV-positive patients diagnosed with syphilis at admission were more likely to be younger, unmarried, and MSM - these factors were associated with a significantly higher risk of syphilis detection in HIV-infected patients.
The COVID-19 pandemic in 2020-2021 led to a decline in the detection of new HIV infections and screening tests in Korea. In this study, the prevalence of syphilis among AIDS patients increased from 2017 to 2022. However, it is unclear whether and how patients in this group could directly contribute to the increased incidence of syphilis in the Korean population. The authors suspect that a shift in the age of AIDS patients toward a younger age could be one such factor. They also state that since younger patients with syphilis were statistically more likely to belong to MSM in this study, this group should receive special care and attention. They emphasise the importance of screening MSM and AIDS patients for other sexually transmitted diseases, including syphilis.
My comments: in Table 1, the marking of values in the table is not entirely consistent and clear - e.g., for categories other than those in which IQR should be in brackets, it might be better to mark (%).
Author Response
We thank the reviewer for taking the time to review our article. We modified Table 1 according to the reviewer’s comment. And we added the following footnote to Table 1.
“ Values in parentheses () are percentages (%), unless otherwise indicated.”
Reviewer 2 Report
Comments and Suggestions for Authors This study analyze the increased cases of incident syphilis in HIV positive patients during the COVID 19 pandemic period in an area of Korea. Althvough the study is well done and the paper well written, in my opinion in the discussion there is a lack of possible explanations for the observed phenomenon (that is, a continuous decline and a sudden increase during the Covid period) I suggest the authors to go more deep in the literature (on HIV+ and HIV- individuals) and to try to add more possible interpretations Once this is done I believe the paper fully deserves to be publishedvAuthor Response
We agreed with the reviewer’s point. So, we added some explanation about the recent increase in Syphilis incidence in HIV-infected patients in the discussion section of our manuscript as below.
"In Korea, the proportion of HIV-infected individuals is less than 0.1% of the total population. Therefore, it is difficult to conclude that those infected with HIV are leading the overall syphilis trend. Rather, the continuous increase in syphilis incidence among HIV-infected individuals during the COVID-19 pandemic may be a reflection of this increase in syphilis incidence in the general population since 2016"